# Involvement of Inwardly Rectifying Potassium (Kir) Channels in the Toxicity of Flonicamid to *Drosophila melanogaster*

**DOI:** 10.3390/insects16010069

**Published:** 2025-01-12

**Authors:** Xuan Liu, Yuying Gao, Tengfei Liu, Hailiang Guo, Jizu Qiao, Jianya Su

**Affiliations:** College of Plant Protection, Nanjing Agricultural University, Nanjing 210095, China; 2020102126@stu.njau.edu.cn (X.L.); gyy18864805832@163.com (Y.G.); 2018102128@stu.njau.edu.cn (T.L.); 2024202079@stu.njau.edu.cn (H.G.); 18734557165@163.com (J.Q.)

**Keywords:** inwardly rectifying potassium channels, flonicamid, flumetnicam, molecular target

## Abstract

Earlier studies confirmed that the pyridineamide insecticide flonicamid inhibits the Kir1 channel in *Nilaparvata lugens* (Stål) (Hemiptera: Delphacidae), thereby reducing insect feeding. This led to the hypothesis that Kir channels are the molecular target of flonicamid. This hypothesis was later contested by researchers who found that flumetnicam, a metabolite of flonicamid, strongly inhibits nicotinamidase (Naam). This inhibition causes nicotinamide accumulation in neurons, impairing the negative geotaxis behavior of *D. melanogaster*. As a result, many peers have now accepted Naam as the target of flonicamid. In our study, we observed that the reduced expression of Kir genes increased tolerance to both flonicamid and its metabolite, suggesting that Kir channels play a significant role in flonicamid’s toxicity to *D. melanogaster*. Additionally, we discovered that flonicamid is more toxic to flies than its metabolite, even though it does not inhibit insect nicotinamidase, raising questions about the assumed role of nicotinamidase as the primary target.

## 1. Introduction

Flonicamid, a pyridineamide insecticide with a distinctive mode of action, was discovered in 1992 by Ishihara Sangyo Kaisha, Ltd., and later developed globally. It is a selective aphicide also used to control leafhoppers, planthoppers, whiteflies, and other pests [1,2,3,4,5,6,7]. Early studies suggested that flonicamid functions as a feeding inhibitor similar to pymetrozine [2]. While pymetrozine was confirmed to target Nan-Iav and Nan-Wtrw channels, neither flonicamid nor 4-trifluoromethylnicotinamide (TFNA-AM), which is a metabolite of flonicamid, interacts with or binds to these channels, indicating a distinct mode of action [8,9,10].

Chordotonal organs are implicated in flonicamid’s toxicity [11,12]. However, flonicamid and TFNA-AM do not act directly on transient receptor potential vanilloid (TRPV) channels [9,10]. Instead, their target appears to be upstream in the mechanosensory signaling cascade that leads to TRPV channel activation [12]. Recent studies show that nicotinamidase (Naam), abundantly expressed in chordotonal neurons, is inhibited by TFNA-AM. This inhibition leads to the accumulation of nicotinamide (NAM) in neurons, impairing the climbing behavior of *Drosophila* [13]. Nicotinamide, an endogenous compound, activates Ca^2+^ channels formed by Nanchung with Wtrw subunits [10] and Nanchung with Wtrw [14]. Feeding aphids with NAM reduces their survival and fecundity, while the foliar application of NAM increases aphid mortality [15]. Based on these findings, flonicamid was classified by the Insecticide Resistance Action Committee (IRAC) in 2023 as a Group 29 insecticide: chordotonal organ nicotinamidase inhibitors [16]. Flonicamid’s toxicity is mediated through its metabolite TFNA-AM, supporting the view that flonicamid acts as a pro-insecticide. In June 2024, the International Organization for Standardization assigned TFNA-AM the common name “flumetnicam” [17].

Noteworthily, flonicamid has been shown to inhibit insect inwardly rectifying potassium (Kir) channels, disrupting salivary gland secretion and the renal tubule excretion of *Nilaparvata lugens* (Stål) (Hemiptera: Delphacidae) in vitro [18], suggesting potential multitarget activity. Kir channels are essential for the establishment of resting membrane potential and the generation, propagation, and regulation of action potentials firing, and they play an important role in the secretory physiology of insects [19]. The disruption of salivary gland and renal tubule functions by Kir inhibitors (e.g., VU625 and VU041) are reportedly lethal to insects and ticks [20,21,22,23,24,25]. Consequently, Kir channels are considered promising targets for novel pesticides [19,21,26,27,28,29,30,31,32]. Thus, the role of Kir channel inhibition in flonicamid toxicity warrants further investigation. This study aimed to clarify the relationship between flonicamid toxicity and Kir channels. We assessed the lethality of flonicamid and flumetnicam in adult *Drosophila* and examined the effects of downregulating Kir1, Kir2, and Kir3 gene expression in the salivary glands both individually and in combination. Additionally, we investigated how Kir expression and flonicamid exposure affect ovary development, revealing a role for Kir channels in reproductive physiology. These findings offer new insights into the toxicological mechanisms of flonicamid and the physiological roles of insect Kir channels, potentially guiding the development of insecticides targeting Kir channels.

## 2. Materials and Methods

### 2.1. Chemicals

Flonicamid (97% technical grade) and its metabolite, flumetnicam (98%) were obtained from Jiangxi Huihe Chemical Co., Ltd., Jiujiang, China and Shanghai Yuanye Biotechnology Co., Ltd., Shanghai, China, respectively. Insecticides were dissolved in acetone as stock solution (50 g/L).

### 2.2. Fly Strains

The w1118 strain of *D. melanogaster* was kindly provided by professor Yufeng Pan (Southeast University, China). The fly strains expressing dsRNA of Kir under control of UAS (v28430 for Kir1, v108140 for Kir2, and v101174 for Kir3) were obtained from the Vienna *Drosophila* Resource Center. Fkh-Gal4 (B78060) for salivary glands expression and Nos-Gal4 (B32179) for ovary expression were from the Bloomington *Drosophila* Stock Center.

UAS-Kir1^dsRNA^ (v28430), UAS-Kir2^dsRNA^ (v108140) and UAS-InvKir3^dsRNA^ (v101174) mated with Gal4 lines separately will lead to the knockdown of Kir expressions in specific tissues, respectively. To achieve the double knockdown of two Kir genes, UAS-InvKir3 and Fkh-Gal4 lines were integrated into one line (Kir3^RNAi^) which express the dsRNA of Kir3 under the control of Fkh-Gal4 in the salivary gland; then, the Kir3^RNAi^ line was mated with UAS-Kir1^dsRNA^ or UAS-Kir2^dsRNA^, respectively, to obtain the double knockdown of Kir1 with *Kir3* or Kir2 with Kir3, respectively. However, the attempt to accomplish the double knockdown of *Kir1* and *Kir2* was unsuccessful due to the embryonic lethality; therefore, the triplex knockdown of Kir1, Kir2 and Kir3 was impossible.

All *D. melanogaster* strains were reared on general purpose medium [33] at 25 °C, humidity of 70 ± 10% and a 12 h:12 h light/dark photoperiod.

### 2.3. Susceptibility Bioassay in D. melanogaster

Stock solutions of flonicamid and flumetnicam were diluted in a liquid diet consisted of 5% sucrose, 2% grape juice, 0.6% propanoic acid and water to prepare insecticide solutions at various concentrations (900–1900 mg/L) in 25 × 95 cm vials. An apparatus was designed in this study to assess the lethal toxicity of flonicamid on adult flies (Figure 1). In this setup, the insecticide was dissolved in liquid diet and absorbed by the cotton bud via capillary action. Adult flies consumed the insecticide while feeding on the cotton bud. The siphon tube with the cotton bud extended slightly above the foam disk, allowing easy access for feeding. Groups of 15 three-day-old male flies were placed in vials and fed the liquid diet during the assays with three replicates for each concentration. The above liquid diet without insecticide served as the negative control. *D. melanogaster* with Kir gene knockdowns in the salivary glands were tested. Two control lines were used: Gal4 line and their corresponding UAS-Kir^dsRNA^ lines.

### 2.4. The Ovariole Number Assay

Based on the preliminary experiment, newly emerged females collected daily of w1118 strain were fed a general purpose medium containing flonicamid (1750 mg/L) for 7 days alongside a negative control diet without insecticide. Ovaries from 7-day-old females were dissected under a stereomicroscope (Zeiss, Jena, Germany) and imaged using a light microscope equipped with a digital video camera (Zeiss, ProgRes 3008 mF, Jenoptik). The number of ovarioles was calculated as the average of mature ovarioles in the left and right ovaries [33]. Each treatment included 12 females with three replicates.

### 2.5. The Fecundity and Hatchablity Assay

Newly emerged flies were fed a general purpose medium containing flonicamid (1750 mg/L) in petri dishes along with a control diet without insecticide. Egg laying was monitored daily for 7 days using groups of 4 pairs of flies. A total of 8 biological replicates were performed. Egg-hatching rates were assessed on a medium of 2% agar with 2% activated carbon.

### 2.6. Statistical Analysis

Dose–response analyses were conducted using Polo-Plus (Version 1.0). LC_50_ values, their 95% fiducial limits (95% FL), slopes, and standard errors (SE) were calculated with significant differences determined by non-overlapping 95% FL. Statistical analyses were performed in GraphPad Prism 7.0 using one-way ANOVA for multiple comparisons and Student’s *t*-test for paired data.

## 3. Results

### 3.1. Knockdown of Kir Genes Reduces Fly’s Susceptibility to Flonicamid and Its Metabolite

The lethal toxicity of flonicamid on w1118 *D. melanogaster* adults was monitored by feeding flies with liquid diet containing flonicamid (900–1900 mg/L) (Figure 1). Mortality rates were recorded daily (Figure 2A). Although flonicamid demonstrated moderate toxicity, a dose-dependent relationship was evident: higher concentrations led to higher mortality. Flonicamid exhibited chronic toxicity with mortality rates increasing gradually over five days and stabilizing by day six. Hence, subsequent assays recorded mortality on the sixth day. Toxicity assays revealed that flonicamid [LC_50_ = 1303.4 (1237.8–1365.7) mg/L] was significantly more toxic than its metabolite, flumetnicam [LC_50_ = 2454.9 (2262.8–2727.7) mg/L] (Figure 2B), suggesting flonicamid has higher insecticidal activity over flumetnicam.

The knockdown of Kir genes in the salivary glands reduced fly susceptibility to flonicamid (Figure 3). The LC_50_ values for flies with Kir1, Kir2, or Kir3 knockdowns [3712.0 (3441.5–4052.4), 4297.7 (3984.0–4755.1), and 2763.5 (2640.2–2920.2) mg/L, respectively] were significantly higher than those of the Fkh-Gal4 control strain [2347.0 (2222.7–2499.4) mg/L] or the UAS-Kir^dsRNA^ parents. Dual knockdowns further increased LC_50_ values; for Kir2 and Kir3 knockdowns, LC_50_ reached 4371.5 (4067.3–4845.5) mg/L, and for Kir1 and Kir3 knockdowns, LC_50_ exceeded 5000 mg/L. No assays were conducted on Kir1 and Kir2 dual knockdowns due to embryonic lethality.

Flumetnicam exhibited lower toxicity than its parent compound, but the knockdown of any Kir gene in the salivary glands substantially reduced fly sensitivity to the metabolite (Figure 4). Simultaneous knockdowns of Kir1 and Kir3 or Kir2 and Kir3 further decreased susceptibility, with LC_50_ values exceeding 5000 mg/L. Thus, Kir knockdowns had comparable effects on the toxicities of both flonicamid and flumetnicam (Figure 4).

### 3.2. Knockdown of Kir Genes Impairs Ovarian Development in D. melanogaster

The impact of Kir channels on ovarian development was assessed in *D. melanogaster*. The knockdown of Kir1 and Kir2 resulted in significantly reduced ovariole numbers (6.4 and 9.0, respectively) compared to the control (18.8) (Figure 5A). Additionally, Kir1 and Kir2 knockdown led to smaller ovaries (Figure 5B). In contrast, Kir3 knockdown had no effect on ovariole number (18.5) or ovary size. These findings suggest that Kir1 and Kir2 are involved in regulating ovarian development, while Kir3 does not appear to influence this process in *D. melanogaster*.

The influence of Kir expression on fecundity and hatchability in *D. melanogaster* was also assessed. The knockdown of Kir1 in the ovary significantly decreased egg production from 86.9 ± 4.2 to 74.1 ± 4.0 (Figure 3A) and hatchability from 87.6% ± 1.6% to 79.8% ± 2.3% (Figure 6D). In contrast, the knockdown of Kir2 reduced hatchability (81.0% ± 1.6%) (Figure 6E), but it did not notably affect egg production (82.4 ± 5.3) (Figure 6B). No significant changes in fecundity (79.6 ± 5.4) (Figure 6C) or hatchability (85.5% ± 1.30%) were observed in Kir3 knockdown flies. These results suggest that egg production and hatchability are regulated by Kir1 channels but not by Kir3 channels in *D. melanogaster*.

### 3.3. Flonicamid Inhibits Ovarian Development in D. melanogaster

Female flies exposed to flonicamid showed significant ovarian development impairment, including a reduced ovariole count (19.24 ± 1.13 vs. 23.69 ± 0.84 for flonicamid vs. control) and smaller ovaries (Figure 7). These findings suggest that flonicamid affects ovarian development by inhibiting Kir channels.

We also examined the effects of flonicamid on fecundity and hatchability in *D. melanogaster*. The results showed a significant decrease in egg production from 88.5 ± 3.2 to 29.5 ± 3.5 (Figure 8A) and hatchability from 80.9% ± 1.70% to 67.0% ± 3.2% (Figure 8B) in flies exposed to flonicamid. These findings indicate that flonicamid inhibits both fecundity and hatchability in *D. melanogaster*. In summary, flonicamid’s effects on fecundity and hatchability resemble those seen with Kir gene knockdown in the ovaries of *D. melanogaster*.

## 4. Discussion

Flonicamid is primarily registered for controlling sap-sucking insects, particularly aphids, and is classified as a selective aphicide [1,2]. Early studies primarily focused on its antifeedant activity in aphids, with its lethal mechanism resembling that of pymetrozine, causing starvation in insects [2,5,6,34]. While flonicamid has been shown to cause uncoordinated locomotion and leg splaying in mosquitoes and locusts, as well as impaired gravitaxis in flies [11,12,13], these effects were observed at high doses and in non-target insects. These findings suggest a potential involvement of chordotonal neurons in flonicamid toxicity [35].

The primary symptom of flonicamid exposure in aphids is feeding inhibition. Studies have shown that the duration of salivation and sap ingestion, measured by electrical penetration graph, is significantly reduced in aphids [*Mzus persicae* (Sulzer) and *Schizaphis graminum* (Rondani) and cotton leafhoppers (*Amrasca biguttula* (Ishida)] when exposed to flonicamid [2,6,36]. Treated insects remain attached to the leaf surface with their proboscis but do not ingest sap. This indicates that flonicamid inhibits salivary secretion and active ingestion, contributing to its toxicity, particularly in aphids. Previous research has demonstrated flonicamid’s inhibitory effects on salivary secretion and renal tubule excretion in planthoppers (*N. lugens*) as well as its inhibition of NlKir1 channels using patch clamp techniques [18]. Kir channels are key regulators of secretion in both salivary and renal systems [19,37]. The inhibition of Kir channels has been shown to significantly reduce salivation in insects like the cotton aphid [22], planthopper [18], mosquito [20,38], horn fly [*Haematobia irritans* (L.)] [24], and lone star tick (*Amblyomma americanum*) [23,39], leading to impaired feeding behavior and subsequent mortality [19]. In the present study, the downregulation of Kir genes in the salivary glands of flies significantly increased their tolerance to flonicamid and its metabolite. The double knockdown of Kir1 with Kir3 or Kir2 with Kir3 further enhanced this tolerance, suggesting that Kir channel disruption plays a role in flonicamid’s toxicological effects.

Although previous reports indicated that flonicamid’s metabolite, TFNA-AM, has strong inhibitory activity, flonicamid itself showed no activity against the Naam enzyme in flies [13]. Interestingly, bioassays conducted on *D. melanogaster* revealed that the parent compound had a significantly higher toxicity (lower LC_50_) than its metabolite (higher LC_50_), contradicting the expected outcome based on Naam inhibition. This suggests that flonicamid and its metabolite may target different molecules or have distinct mechanisms of action in various insects, supporting the idea that flonicamid may be a multitarget insecticide.

The increased tolerance to flonicamid in flies with salivary gland-specific knockdown of Kir genes supports the involvement of Kir channels in regulating salivary secretion. The study also showed that Kir1 and Kir2 downregulation affected ovarian development and egg production, while Kir3 did not. The double knockdown of Kir1 and Kir2 resulted in embryonic lethality, whereas double knockdown with Kir3 did not, indicating functional complementarity between Kir1 and Kir2 and differences in function between Kir1/Kir2 and Kir3 [40,41]. And the enhanced tolerance with the double knockdown of Kir1 with Kir3 or Kir2 with Kir3 suggests the inhibitory activities of flonicamid on these Kir channels. It was also demonstrated that Kir1 and Kir2 are involved in the regulation of fly’s embryonic development, while Kir3 does not participate in this process. Disrupting the function of Kir channels using flonicamid produces effects like the knockdown expression of Kir genes, severely impairing fly’s embryonic development. These findings suggest that Kir channels, particularly Kir1 and Kir2, play essential roles in secretion, reproduction and embryonic development in *D. melanogaster*, aligning with similar observations in *A. aegypti* mosquitoes [42]. Given that flonicamid inhibits Kir channels and affects insect reproduction, it highlights the potential for developing novel insecticides targeting Kir channels.

## 5. Conclusions

This study demonstrates that the downregulation of Kir genes significantly enhances fly tolerance to flonicamid and its metabolite. Flonicamid exhibited higher toxicity against flies than its metabolite, which contradicts previous findings that flonicamid has no activity against Naam. Combined with the existing literature, these results suggest that flonicamid and its metabolite may act on multiple targets with Kir channels being a key molecular target. These findings offer valuable insight into the toxicological mechanisms of flonicamid and may guide the development of insecticides with novel modes of action. However, further research is needed to fully elucidate the contribution of different molecular targets to the insecticidal effects of flonicamid.

## Figures and Tables

**Figure 1 insects-16-00069-f001:**
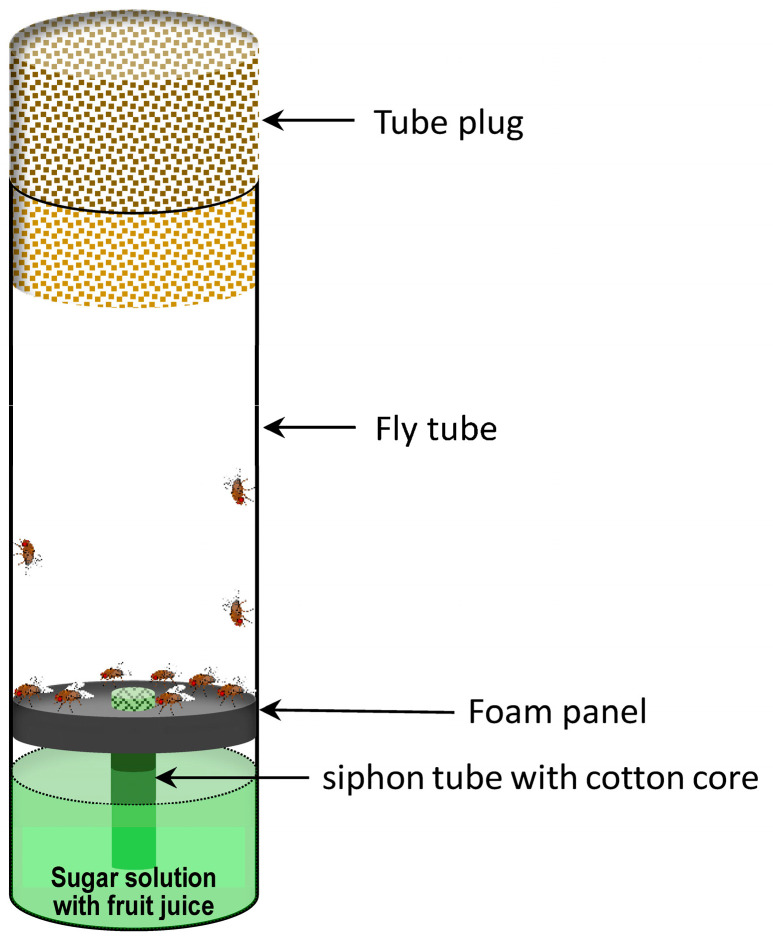
The apparatus for toxicity assay of flonicamid on fly adults.

**Figure 2 insects-16-00069-f002:**
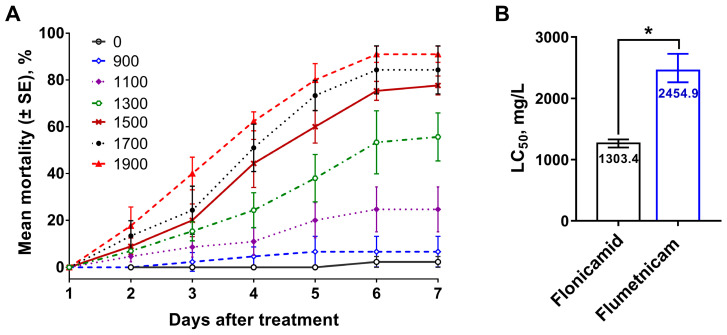
The toxicities of flonicamid and its metabolite to w1118 fly. (**A**) The time course changes in mortalities of w1118 fly exposed to flonicamid. (**B**) The comparison of lethal toxicities between flonicamid and its metabolite. Asterisk represents significant difference between treatments (*p* < 0.05).

**Figure 3 insects-16-00069-f003:**
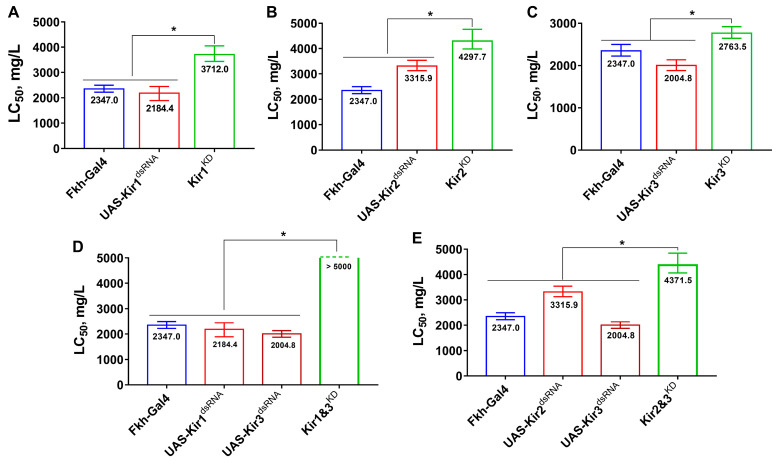
The susceptibilities of *D. melanogaster* with Kir knockdown in salivary glands to flonicamid. (**A**–**C**) represent the susceptibilities of fly with knockdown of Kir1, Kir2, and Kir3, respectively. (**D**) represents the susceptibilities with double knockdown of Kir1 and Kir3. (**E**) represents the susceptibilities with double knockdown of Kir2 and Kir3. Asterisks represent significant differences (*p* < 0.05).

**Figure 4 insects-16-00069-f004:**
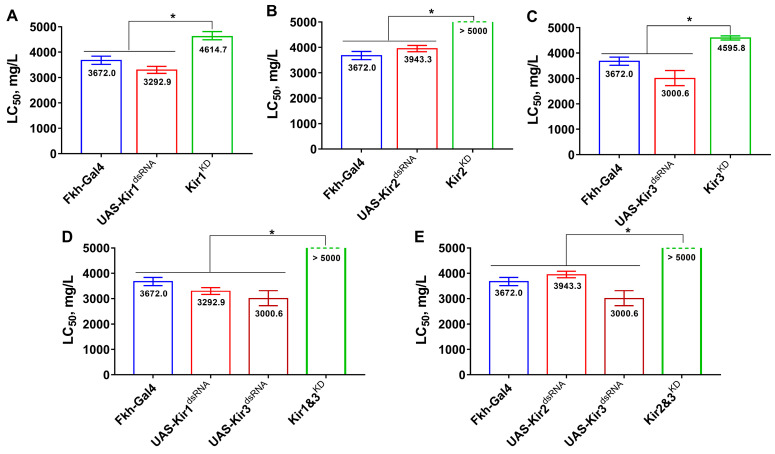
The susceptibilities of *D. melanogaster* with Kir knockdown in salivary glands to flumetnicam. (**A**–**C**) represent fly’s susceptibilities after knockdown of Kir1, Kir2, and Kir3, respectively. (**D**) represents the susceptibilities with double knockdown of Kir1 and Kir3. (**E**) represents the susceptibilities with double knockdown of Kir2 and Kir3. Asterisks represent significant difference (*p* < 0.05).

**Figure 5 insects-16-00069-f005:**
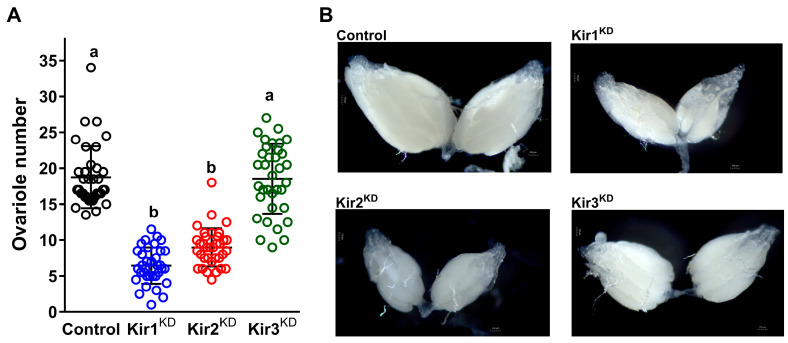
Effects of Kir channels on ovarian development in *D. melanogaster*. (**A**) Knockdown of Kir genes significantly reduced ovariole numbers compared to the control group with statistical differences indicated by different letters (one-way ANOVA with Dunnett’s test, *p* < 0.0001). The F1 offsprings of Nos-Gal4 and w1118 were used as the control. (**B**) Knockdown of Kir genes led to a marked decrease in ovary size. Scale bar: 100 μm. The F1 offsprings of nanos-Gal4 and w1118 were used as the control.

**Figure 6 insects-16-00069-f006:**
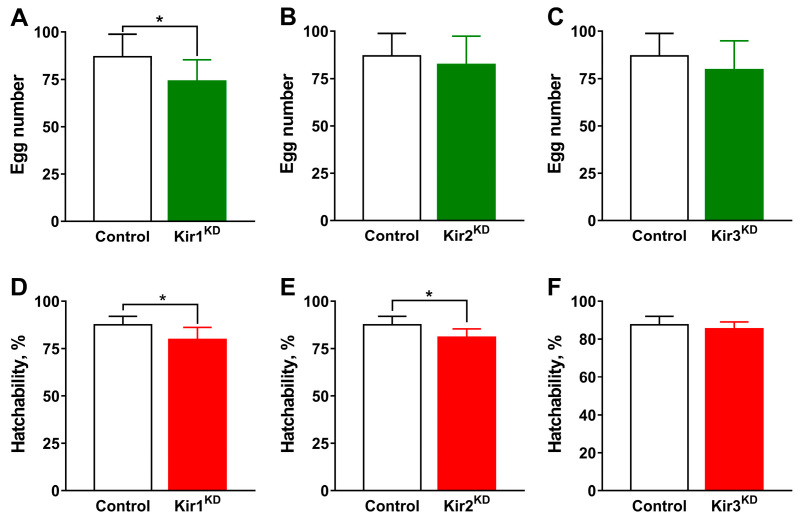
Effects of Kir expression on egg production and hatchability in *D. melanogaster*. (**A**–**C**) show the effects on egg production following knockdown of Kir1, Kir2, and Kir3, respectively. (**D**–**F**) depict the effects on hatchability following knockdown of Kir1, Kir2, and Kir3, respectively. The F1 offsprings of Nos-Gal4 and w1118 were used as the control (CK). Asterisk indicates significant difference (Student’s *t*-test, *p* < 0.05).

**Figure 7 insects-16-00069-f007:**
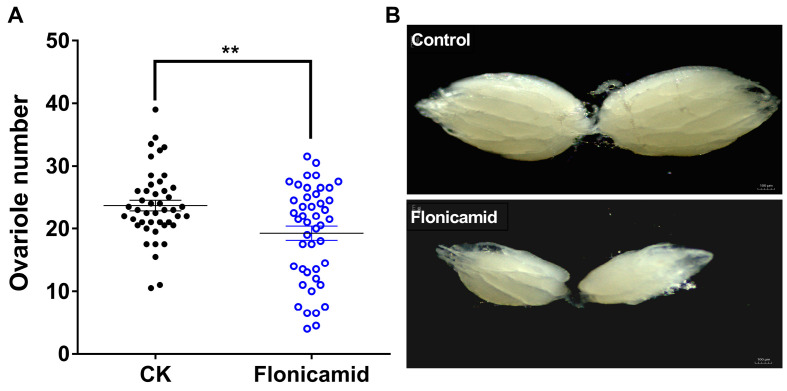
Effect of flonicamid on ovarian development in *D. melanogaster*. (**A**) Effect of flonicamid on ovarole numbers. ** indicates significant differences (Student’s *t*-test, *p* < 0.01). (**B**) Effect of flonicamid on ovary size compared to the control (w1118). Scale bar: 100 μm.

**Figure 8 insects-16-00069-f008:**
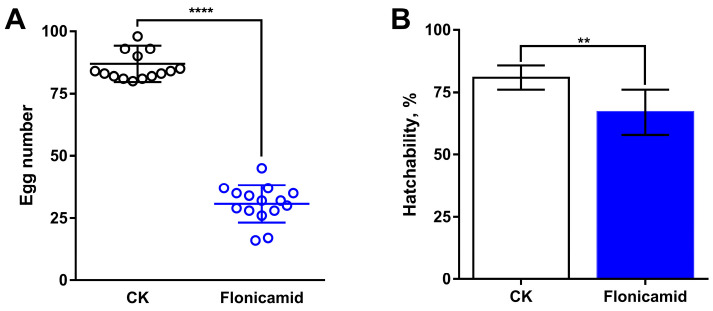
Effect of flonicamid on egg production (**A**) and hatchability (**B**) in *D. melanogaster*. ** and **** represent significant difference at the levels of 0.01 and 0.0001 (Student’s *t*-test), respectively. The horizonal lines represent the mean and their 95% FLs, respectively.

## Data Availability

All data analyzed in this study are included in this published article.

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
