# Peer review of "Involvement of Inwardly Rectifying Potassium (Kir) Channels in the Toxicity of Flonicamid to Drosophila melanogaster"

_insects, 2025, doi:10.3390/insects16010069_

Round 1
Reviewer 1 Report
Comments and Suggestions for Authors
The manuscript "Involvement of Inward Rectifier Potassium (Kir) Channels in the Toxicity of Flonicamid to Drosophila melanogaster". The topic is interesting enough to investigate and the experiments are well-designed. The effects of flonicamid to Drosophila melanogaste were evaluated, and the result has offered new insights into the physiological roles of insect Kir channels and the toxicological mechanisms of flonicamid. There were, however, a few issues in the manuscript that should be modified as described below.
1. The abbreviations such as TRPV, IRAC, ISO, that appear for the first time need to be supplemented with the full name
2. Line 39, a short brief about “4-trifluoromethylnicotinamide” should supplied.
3. Line 73, Are flonicamid and flumetnicam prodrugs or commercial drugs? Please indicate the specific concentration and dosage form.
4. Line 74, a space is required between “Ltd.and”
5. Line 70, the “is” should be “are”.
6. Line 91, supply the reference of “standard fly diet”.
7. Line 96, supply the range of “various concentrations”
8. Line 109, which strain are these newly emerged females from? And also what is the reason for choosing this treatment concentration (1750 mg/L) and time (7 days)?
9. In figure 5, which fly strain is the control? It was the same as in figure7?
10. In figure 5 and 7, It would be good to have the scale bar labelled in the images to facilitate better comparison and perception of the results by the reader
11. Line 114, “more than 10 females” is an inaccurate description, and the number of specific dissections per treatment group needs to be stated.
12. Line 162, 168, 196, the “p” should be in italics, please check the manuscript
13. In figure7B, the abbreviation of control should be CK.
14. The result showed that flonicamid decreases the hatchability of eggs, suggesting some effect on the normal physiology and biochemistry of eggs or embryonic development. This needs to be discussed and illustrated in the discussion section in conjunction with the relevant literature.
15. The formatting of the references does not meet the requirements of the journal and needs to be carefully revised.
Author Response
1. The abbreviations such as TRPV, IRAC, ISO, that appear for the first time need to be supplemented with the full name
Reponse: Done.
2. Line 39, a short brief about “4-trifluoromethylnicotinamide” should supplied.
Response: A brief explanation for 4-trifluoromethylnicotinamid has added in the text.
3. Line 73, Are flonicamid and flumetnicam prodrugs or commercial drugs? Please indicate the specific concentration and dosage form
Response: Add the concentration of technical grade, flonicamid 97% and flumetnicam 98%
4. Line 74, a space is required between “Ltd.and”
Response: Done
5. Line 70, the “is” should be “are”.
Response: Done
6. Line 91, supply the reference of “standard fly diet”.
Response: a reference has been added.
7. Line 96, supply the range of “various concentrations”
Response: The concentrate range is added.
8. Line 109, which strain are these newly emerged females from? And also what is the reason for choosing this treatment concentration (1750 mg/L) and time (7 days)?
Response: w1118 strain was used in this study. The concentration (1750 mg/L) and time (7 days) was based on our premilinary experiments.
9. In figure 5, which fly strain is the control? It was the same as in figure7?
Response: The F1 offsprings of nanos-Gal4 and w1118 were used as the control in Figure 5, w1118 was used as control in Figure 7. The details have added in the figure legend.
10. In figure 5 and 7, It would be good to have the scale bar labelled in the images to facilitate better comparison and perception of the results by the reader
Response: The scale bar has been labelled in these original figures
11. Line 114, “more than 10 females” is an inaccurate description, and the number of specific dissections per treatment group needs to be stated.
Response: The number of dissections (12 females) was specified and revised.
12. Line 162, 168, 196, the “p” should be in italics, please check the manuscript
Response: Done.
13. In figure7B, the abbreviation of control should be CK.
Response: Done
14. The result showed that flonicamid decreases the hatchability of eggs, suggesting some effect on the normal physiology and biochemistry of eggs or embryonic development. This needs to be discussed and illustrated in the discussion section in conjunction with the relevant literature.
Response: Done
15. The formatting of the references does not meet the requirements of the journal and needs to be carefully revised.
Response: Done
Reviewer 2 Report
Comments and Suggestions for Authors
This is a very interesting study. The English language is excellent, and the overall structure and presentation are coherent and well-organized. I have some suggestions pointed out below.
Simple abstract: Please modify the simple abstract so it can be conceivable by a broader audience. For instance, explain what flonicamid, nicotinamidase, Drosophila or Nilaparvata lugens are (order:family). The authors may introduce short explanations like “ flumetnicam, a metabolite of the pyridineamide insecticide flonicamid…”.
Lines 14, 29-30: This may be considered an exaggeration. Multiple studies have examined the role of Kir channels to the toxicity of flonicamid. I would suggest focusing on the effects of flonicamid in D. melanogaster.
Lines 63-66: Please avoid presenting the results in the introduction section.
Line 77: Please mention the full name and authorities of D. melanogaster along with the strain since this is the first time the species has been mentioned in the text.
Line 78: Please mention the Upstream Activating Sequence next to its first abbreviated mention.
Lines 77-81: Please consider elaborating on the duration and conditions in which the flies were maintained in each facility.
Line 96: Please refer to the concentrations prepared
Lines 109-110: Were the newly emerged flies collected daily or one-day old flies are considered newly emerged by the authors? Also, please consider adding a reference to support the ovariole maturity at seven days of age.
Line 114: “more than 10 females”? Please include the exact number for each replication or the total number of females used.
Line 117: Do the authors mean egg-laying?
Line 126: I would suggest removing the denotation “p < 0.05 (*), p < 0.01 (**), and p < 0.001 (***)” and including it in the figure legends.
Lines 129, 169, 198: I would suggest refraining from presenting the results in the subsections. The authors may use titles such as “Ovarian development in D. melanogaster” instead of “Flonicamid inhibits ovarian development in D. melanogaster”
Line 164: Please correct the typing error “Drosophil”. Also, please consider using “D. melanogaster” instead of just “Drosophila”.
Figures 2, 5,6,7: Please, explain in the legend if “CK” represents controls or what it stands for.
Lines 228-231: Please add a reference here.
Line 230: Please correct the latin name to Myzus persicae and add the authorities of the latin names throughout the text.
Lines 243-244: It is already known that Kir channel disruption plays a role in flonicamid's toxicological effects, but how do the authors comment on the results of the enhanced tolerance with the double knockdown of Kir1 with Kir3 or Kir2 with Kir3?
Discussion: I would suggest elaborating a little more on the results of this study. For instance, how do the authors believe that salivary secretion, ovarian development, and egg production regulation compensate for the high tolerance?
Author Response
Simple abstract: Please modify the simple abstract so it can be conceivable by a broader audience. For instance, explain what flonicamid, nicotinamidase, Drosophila or Nilaparvata lugens are (order:family). The authors may introduce short explanations like “ flumetnicam, a metabolite of the pyridineamide insecticide flonicamid…”.
Response: Done
Lines 14, 29-30: This may be considered an exaggeration. Multiple studies have examined the role of Kir channels to the toxicity of flonicamid. I would suggest focusing on the effects of flonicamid in D. melanogaster.
Response: Done
Lines 63-66: Please avoid presenting the results in the introduction section.
Response: We delete the results in introduction section.
Line 77: Please mention the full name and authorities of D. melanogaster along with the strain since this is the first time the species has been mentioned in the text.
Response: Done
Line 78: Please mention the Upstream Activating Sequence next to its first abbreviated mention.
Response: The UAS-Gal4 binary expression system has become a fundamental tool in Drosophila research, and its basic principles no longer require detailed introduction in most contemporary publications.
Lines 77-81: Please consider elaborating on the duration and conditions in which the flies were maintained in each facility.
Response: The cuture conditions had been described in the end of this section. We add the relative reference.
Line 96: Please refer to the concentrations prepared
Response: Done
Lines 109-110: Were the newly emerged flies collected daily or one-day old flies are considered newly emerged by the authors? Also, please consider adding a reference to support the ovariole maturity at seven days of age.
Response: Revised, and a reference was added.
Line 114: “more than 10 females”? Please include the exact number for each replication or the total number of females used.
Response: Revised
Line 117: Do the authors mean egg-laying?
Response: Egg-laying
Line 126: I would suggest removing the denotation “p < 0.05 (*), p < 0.01 (**), and p < 0.001 (***)” and including it in the figure legends.
Response: Done
Lines 129, 169, 198: I would suggest refraining from presenting the results in the subsections. The authors may use titles such as “Ovarian development in D. melanogaster” instead of “Flonicamid inhibits ovarian development in D. melanogaster”
Response: We believes that the original subheadings better reflect the structure and hierarchy of the paper.
Line 164: Please correct the typing error “Drosophil”. Also, please consider using “D. melanogaster” instead of just “Drosophila”.
Response: Done
Figures 2, 5,6,7: Please, explain in the legend if “CK” represents controls or what it stands for.
Response: Added the explanation for CK
Lines 228-231: Please add a reference here.
Response: Done
Line 230: Please correct the latin name to Myzus persicae and add the authorities of the latin names throughout the text.
Response: Done
Lines 243-244: It is already known that Kir channel disruption plays a role in flonicamid's toxicological effects, but how do the authors comment on the results of the enhanced tolerance with the double knockdown of Kir1 with Kir3 or Kir2 with Kir3?
Response: Done
Discussion: I would suggest elaborating a little more on the results of this study. For instance, how do the authors believe that salivary secretion, ovarian development, and egg production regulation compensate for the high tolerance?
Response: Our results did not support the compensation of ovarian development and egg production regulation for the high tolerance to flonicamid. Therefore, we did not discuss this issue.
Reviewer 3 Report
Comments and Suggestions for Authors
Interestingly, Gao et al. investigated the involvement of inwardly rectifying potassium (Kir) channels in the toxicity of flonicamid against Drosophila melanogaster. Therein, the biochemical-genetic approach has been combined effectively to dispute the generally accepted view that flonicamid has nicotinamidase, Naam, as its prime molecular target. The innovative use of gene knockdown techniques to decipher Kir's role in salivary gland and reproductive physiology is one of the many strong points. Though generally well-written, there are areas for improvement. For instance, a discussion on possible off-target effects is lacking, and the wider ecological implications of flonicamid application have not been fully assessed. Language quality is 8/10: minor grammatical discrepancies and occasional opacity. Overall, the scientific soundness and relevance of this study deserve a score of 85/100.
Major points:
The experimental design is robust. The gene knockdown techniques are accurate, and the methodology uses a dose-response type assay for toxicity. However, controls for specificity to exclude possible off-target RNAi effects are lacking.
Given the critical importance of statistics, some of the findings, such as the differences in LC50 values between flonicamid and flumetnicam, lack mechanistic explanation.
The discussion on the different functions of Kir1, Kir2, and Kir3 is interesting, but the possibility of compensatory mechanisms or redundancy among subtypes of Kir is not discussed.
Although results showing embryonic lethality in the case of Kir1 and Kir2 double knockdown are interesting, this is poorly contextualized regarding the larger physiological implications.
Although the figures do a good job of presenting the results, they could use some annotation to highlight important information, such as notable variations in ovary sizes.
Although the authors could have postulated a multitarget activity for flonicamid, this was not proven, and therefore there is some uncertainty regarding the exact mode of action.
There is little discussion of the ecological and practical issues in relation to possible resistance development.
No resoaning was given to the very high concentrations of flonicamid used in experimental setups.
Minor points:
In the abstract, the claim of providing "new insights" into flonicamid toxicity might be better supported by specific examples.
A better introduction should put Kir channels in a broader perspective than just salivary and reproductive functions within insect physiology.
Supporting methodological information, such as justification for the choice of specific knockdown constructs, is either lacking or needs to be expanded.
While extensive references to previous works exist, some parts of the discussion are repetitive.
The obtained results demonstrate conclusively that Kir channels have a critical role in flonicamid toxicity, beyond the reigning Naam-centric view. Results obtained so far point to the possibility of developing new insecticides with new modes of action against Kir channels. Poor exploration of the resistance risks and off-target effects is the limiting factor for the applicability of results.
The present work is an important contribution to the field of insect physiology and toxicology, in which Kir channels have been given new relevance. Although well-conceived, the study has certain limitations that, once addressed, will strengthen its impact. I recommend its publication following minor revisions.
Author Response
Major points:
The experimental design is robust. The gene knockdown techniques are accurate, and the methodology uses a dose-response type assay for toxicity. However, controls for specificity to exclude possible off-target RNAi effects are lacking.
Response: The UAS-Gal4 binary expression system has become a fundamental tool in Drosophila research, the RNAi fly used in this study was obtained from Vienna Drosophila Resource Center. The inverted repeat of specific gene for RNAi in this fly center was designed based on the genomic sequence of fly and the possibility of off-target RNAi effects had been excluded when the sequence was designed.
Given the critical importance of statistics, some of the findings, such as the differences in LC50 values between flonicamid and flumetnicam, lack mechanistic explanation.
Response: Done
The discussion on the different functions of Kir1, Kir2, and Kir3 is interesting, but the possibility of compensatory mechanisms or redundancy among subtypes of Kir is not discussed.
Response: We believe the compensatory mechanisms or redundancy among subtypes of Kir is not the focus of the paper and therefore have not discussed it in detail.
Although results showing embryonic lethality in the case of Kir1 and Kir2 double knockdown are interesting, this is poorly contextualized regarding the larger physiological implications.
Response: The focus of this study is to demonstrate that the insecticidal activity of flonicamid is related to Kir channels. Accordingly, the discussion primarily revolves around this key issue. As a result, the embryonic lethality caused by the double knockdown of Kir1 and Kir2 was not addressed in detail.
Although the figures do a good job of presenting the results, they could use some annotation to highlight important information, such as notable variations in ovary sizes.
Response: Due to the significant decrease in the number of ovarian tubules observed, we focused on statistical analysis of the changes in the number of ovarian tubules.
Although the authors could have postulated a multitarget activity for flonicamid, this was not proven, and therefore there is some uncertainty regarding the exact mode of action.
Response: This is only our speculation, and the true target of flonicamid remains to be determined through further research.
There is little discussion of the ecological and practical issues in relation to possible resistance development.
Response: This paper does not address the issue of insecticide resistance, and therefore it is not discussed.
No resoaning was given to the very high concentrations of flonicamid used in experimental setups.
Response: In instruction section we had mentioned that fly is the non-target insect.
Minor points:
In the abstract, the claim of providing "new insights" into flonicamid toxicity might be better supported by specific examples.
Response: Done
A better introduction should put Kir channels in a broader perspective than just salivary and reproductive functions within insect physiology.
Response: Done
Supporting methodological information, such as justification for the choice of specific knockdown constructs, is either lacking or needs to be expanded.
Response: The UAS-Gal4 binary expression system has become a fundamental tool in Drosophila research, and its basic principles no longer require detailed introduction in most contemporary publications.
While extensive references to previous works exist, some parts of the discussion are repetitive.
Response: We do not believe that the discussion section is repetitive or redundant. These discussions are essential for clarifying whether the toxicological effects of flonicamid involve Kir channels. Removing certain parts may lead to an unclear explanation of the issues.
The obtained results demonstrate conclusively that Kir channels have a critical role in flonicamid toxicity, beyond the reigning Naam-centric view. Results obtained so far point to the possibility of developing new insecticides with new modes of action against Kir channels. Poor exploration of the resistance risks and off-target effects is the limiting factor for the applicability of results.
Response: The focus of this paper is to elucidate the relationship between Kir channels and the toxicity of flonicamid. The issues of resistance risk and off-target effects are not addressed in this study, and therefore, are not discussed.